# Imidazole Processing of Wheat Straw and Eucalyptus Residues—Comparison of Pre-Treatment Conditions and Their Influence on Enzymatic Hydrolysis

**DOI:** 10.3390/molecules26247591

**Published:** 2021-12-15

**Authors:** Pedro M. A. Pereira, Joana R. Bernardo, Luisa Bivar Roseiro, Francisco Gírio, Rafał M. Łukasik

**Affiliations:** Laboratório Nacional de Energia e Geologia, I. P., Unidade de Bioenergia e Biorrefinerias, Estrada do Paço do Lumiar 22, 1649-038 Lisbon, Portugal; pedro18broca@gmail.com (P.M.A.P.); joana-bernardo@bioref-colab.pt (J.R.B.); luisa.roseiro@lneg.pt (L.B.R.); francisco.girio@lneg.pt (F.G.)

**Keywords:** pre-treatment, imidazole, hardwood, biomass, biorefinery

## Abstract

Biomass pre-treatment is a key step in achieving the economic competitiveness of biomass conversion. In the present work, an imidazole pre-treatment process was performed and evaluated using wheat straw and eucalyptus residues as model feedstocks for agriculture and forest-origin biomasses, respectively. Results showed that imidazole is an efficient pre-treatment agent; however, better results were obtained for wheat straw due to the recalcitrant behavior of eucalyptus residues. The temperature had a stronger effect than time on wheat straw pre-treatment but at 160 °C and 4 h, similar results were obtained for cellulose and hemicellulose content from both biomasses (ca. 54% and 24%, respectively). Lignin content in the pre-treated solid was higher for eucalyptus residues (16% vs. 4%), as expected. Enzymatic hydrolysis, applied to both biomasses after different pre-treatments, revealed that results improved with increasing temperature/time for wheat straw. However, these conditions had no influence on the results for eucalyptus residues, with very low glucan to glucose enzymatic hydrolysis yield (93% for wheat straw vs. 40% for eucalyptus residues). Imidazole can therefore be considered as a suitable solvent for herbaceous biomass pre-treatment.

## 1. Introduction

There is a growing demand for solutions providing integration and flexibility in the European energy system. These solutions should create flexibility between intermittent electricity and sustainable fuel production and, at the same time, enable production under economically competitive conditions from alternative carbon sources [1,2,3,4]. To accomplish this objective, it is essential to develop flexible, selective, robust and less energy-demanding integrated pre-treatments with the simultaneous separation of contaminants, as well as optimization of pre-treatment technologies compatible with the use of multiple feedstocks. This includes the efficient utilization of resources to produce purified fractions of carbohydrates, lignin and other compounds for further processing and conversion into a wide spectrum of products, such as proteins, biopolymers, organic acids, and furfural and its derivatives [5,6,7]. 

The selection of the pre-treatment method depends on the biomass type as well as the desired product, e.g., the delivery of upgradable sugars or lignin-derived commodities. Several pre-treatment technologies are currently employed to overcome the recalcitrance of lignocellulose, to increase hydrolysis efficiency and to improve the yields to monomeric sugars [8,9,10,11]. Among them are mostly chemical methods, e.g., with acids or alkali. Nevertheless, novel biomass pre-treatment protocols, such as those based on non-hazardous catalysts and/or green solvents that simultaneously enable a reduction in the number of hydrolytic enzymes needed for cellulose hydrolysis and for the fermentation inhibitors, are still required. At the same time, these are the core options to obtain lignin (or its derivatives) and pure sugar streams [12,13,14,15]. Subsequently, these streams can work as feedstock for fermentation into biofuels and other value-added applications. 

Imidazole, along with ionic liquids [16,17,18,19], high-pressure fluids [20,21,22], and deep eutectic solvents [23,24], belongs to these new pre-treatment options and, up until now, has demonstrated considerable potential in the valorization of biomass, especially in the context of biorefinery [4] focused on value-added products. Imidazole, being an environmentally benign and non-hazardous solvent, turns the process of biomass pre-treatment into an attractive alternative, offering the possibility of delivering a depolymerized lignin and highly hydrolyzable polysaccharide fraction [25,26]. As such, the imidazole processing of biomass is similar to that presented by organosolv pre-treatment [27,28].

The present work describes the application of imidazole pre-treatment in two representative feedstocks of lignocellulosic residues, one of agricultural origin (wheat straw) and the other of forest origin (eucalyptus residues). Wheat straw and eucalyptus residues are two major lignocellulosic feedstocks for sustainable bioeconomic development in Europe. Subsequently, pre-treated solids of both biomasses rich in polysaccharides are scrutinized for the ability to originate concentrated reducing sugar streams, obtained in the enzymatic saccharification process.

## 2. Experimental Section

### 2.1. Materials

The wheat straw sample was delivered by ECN (Energy Research Center of the Netherlands), from The Netherlands. The eucalyptus residues were kindly provided by The Navigator Company from their papermill in Cacia, Portugal. Wheat straw and eucalyptus residues moisture contents were found to be 9.8 and 8.4 wt %, respectively, and were determined using an AMB-50 moisture analyzer (Adam Equipment Inc., Oxford CT, USA).

Both feedstocks were ground with a knife mill, IKA^®^ WERKE, MF 10 basic (Staufen, Germany), into particles smaller than 0.5 mm, homogenized in a defined lot, and stored in plastic containers at room temperature prior to further use. 

For the pre-treatment experiments and post-reaction processing, the following reagents were used: imidazole with a purity of 99% w/w, purchased from Alfa Aesar, (Karlsruhe, Germany), NaOH pellets (99% purity) supplied by Eka Chemicals/Akzonobel (Bohus/Sweden); 25% HCl aqueous solution prepared from 37% HCl solution (VWR Chemicals/AnalaR NORMAPUR^®^, Alfragide, Portugal)) using ultrapure water (18.2 MΩ/cm) produced via the PURELAB Classic of Elga system; HCl solution with a pH of 2, prepared from distilled water and a 25% HCl solution; 96% ethanol (v/v) and acetonitrile at 99.9% purity, both from the Carlo Erba Group, Aresa, Italy. For filtration, cloth (made from a cotton shirt), paper (ø = 150 mm, n. 1238, acquired from Filter-Lab^®^, Filtros Anoia, S.A. Barcelona, Spain) and nylon filters (0.45 µm, also from Filter-Lab^®^, Filtros Anoia, S.A. Barcelona, Spain) were used. 

Glucose (≥98 wt %, Merck, Darmstadt, Germany), xylose (≥98 wt %, Merck, Germany), arabinose (≥98 wt %, Merck, Germany), furfural (99 wt %, Sigma-Aldrich, Germany), 5-hydroxymethylfurfural (99 wt %, Sigma-Aldrich, Taufkirschen, Germany) and acetic acid (glacial, 99.8 wt %, Merck, Darmstadt, Germany) were used for the qualitative and quantitative HPLC analyses of the obtained liquids and solids. Sulfuric acid (96 wt %, Panreac, Barcelona, Spain) was used to prepare the mobile phase for the HPLC analyses (5 mM sulfuric acid).

For the enzymatic hydrolysis assays, 0.1 M sodium citrate buffer (pH 4.8) prepared from citric acid monohydrate (99.7% purity) and tris-sodium citrate (>99% purity), both from VWR International Ltd. (Leicester, England), and 2 wt % sodium azide solution were used. Celli^®^CTec2 (Cellulase, enzyme blend) solution, kindly provided by Novozymes Europe, Denmark, was employed in the enzymatic reaction. 

### 2.2. Biomass and Pre-Treated Solid Characterization

Both biomasses and pre-treated solids were characterized according to the NREL method [29], to determine the total moisture, total lignin and polysaccharide contents. Acid-insoluble lignin was determined gravimetrically, while the acid-soluble lignin content was established using UV spectrophotometry. The content of glucan and hemicelluloses (xylan, arabinan, and acetyl groups) was determined using high-performance liquid chromatography (HPLC). Furthermore, for native biomasses, total extractives, ash and protein contents were determined according to the standard methods—NREL/TP-510-42619 [30], NREL/TP-510-42622 [31] and ISO 8968-1:2014 [32], respectively. All analyses were conducted in duplicate and are presented as mean values. 

### 2.3. Biomass Processing 

The pre-treatment of biomass with imidazole was carried out on the basis of a previously developed method presented elsewhere [15] and is outlined in Figure 1. For all the experiments, 5 g of air-dried biomass and 45 g of imidazole were placed into a 100 mL Schott flask. The reaction vessel was then placed in an oil bath to guarantee continuous and controlled stirring and heating. Reaction time started from the moment when the desired temperature was reached. The reaction temperature and time were in the range of 130–160 °C and 2 to 4 h. When the process ended, the flask containing the mixture was removed from the bath and cooled down to 90 °C; then, 90 mL distilled water was added slowly. Next, the obtained mixture was transferred to a 500 mL Erlenmeyer flask and stirred for 1 h. This step promoted the precipitation of pulp, which was collected and oven-dried at 45 °C for 24 h.

### 2.4. Enzymatic Hydrolysis of Solids 

Enzymatic hydrolysis assays were performed to evaluate the effect of pre-treatment on glucan and xylan conversion to their corresponding monomers. The adopted procedure was based on the standard NREL protocol [33]. For this purpose, native biomasses and pre-treated materials were subjected to enzymatic hydrolysis at a 2% w/v total solids (dry weight basis) concentration in 50 mL vials, with 5 mL of 0.05 M sodium citrate buffer (pH of 5), prepared from citric acid monohydrate and tris-sodium citrate, and 100 μL of a 2 wt % sodium azide solution to prevent the undesired growth of microorganisms. Distilled water was added to reach the 5.0 mL total volume, taking into account the volume of enzyme added. The enzyme loading was 60 FPU/g glucan of Celli^®^CTec2 (199.9 FPU/mL). The enzymatic hydrolyses were performed in a shaking incubator (Optic Ivymen system—Madrid, Spain) at 180 rpm and 50 °C. Reaction blanks for both the substrate and the enzyme solution were carried out under the same experimental conditions. 

At the desired reaction time points (1, 4, 6, 24, 48 and 72 h), a 0.1 mL sample was taken, diluted, filtered through a 0.45 mm filter and analyzed via HPLC. All enzymatic hydrolysis assays were performed in duplicate.

The glucose and xylose yields were calculated according to the following formulas: $\mathrm{Glucose}\;\mathrm{yield}\;\left(\%\right)=\frac{\frac{162}{180}\;\times \;\left[\mathrm{glucose}\right]\times V}{x_{\mathrm{cellulose}-\mathrm{rich}\;\mathrm{solid}}\;\times \;\mathrm{glucan}\;\mathrm{content}}\;\times \;100$ and $\mathrm{Xylose}\;\mathrm{yield}\left(\%\right)=\frac{\frac{132}{150}\;\times \;\left[\mathrm{xylose}\right]\times V}{x_{\mathrm{cellulose}-\mathrm{rich}\;\mathrm{solid}}\;\times \;\mathrm{xylan}\;\mathrm{content}}\;\times \;100$, where the [glucose] and [xylose] concentrations were measured in g/L obtained in the hydrolysate, the values $\frac{162}{180}$ and $\frac{132}{150}$ are the dehydration factors, V is the volume of solution in L, $x_{\mathrm{cellulose}-\mathrm{rich}\;\mathrm{solid}}$ refers to the cellulose-rich fraction in dry weight biomass, used in the enzymatic hydrolysis and expressed in g, and glucan or xylan contents refer to their contents in the solid fraction used for the enzymatic hydrolysis.

### 2.5. Chemical Analysis

HPLC Analysis

The liquid phases obtained from the pre-treatment of biomass with imidazole and enzymatic hydrolyses were analyzed using an Agilent 1100 series machine with a Bio-Rad Aminex HPX-87H column (Bio-Rad, Hercules, CA, USA). The liquids obtained from the characterization of native and pre-treated biomass were also analyzed by HPLC. Analyses were performed at 65 °C, with 5 mmol/L H_2_SO_4_ used as the mobile phase, at a flow rate of 0.6 mL/min. The detection was performed using an RID (refractive index detector) for monosaccharides (glucose, xylose and arabinose) and acetic acid, and a DAD (diode array detector) at 280 nm wavelength for furans (furfural (furan-2-carbaldehyde) and 5-HMF ≡ 5-hydroxymethylfurfural (5-(hydroxymethyl)-2-furaldehyde)). The quantitative analyses were performed using the external calibration method with standard solutions.

## 3. Results and Discussion

### 3.1. Biomass Composition

The composition of both biomasses in their native form was determined, and the obtained results are summarized in Table 1. 

The obtained results demonstrate that the eucalyptus residues contain almost 20% more cellulose than wheat straw. However, in the case of cellulose and hemicellulose, the content of these polysaccharide fractions was very similar for both biomasses, above 60% that of native feedstocks. On the other hand, the difference in lignin content in both biomasses was more noticeable. Eucalyptus residues contained twice the lignin of wheat straw. Regarding ash, wheat straw was significantly richer in ash than eucalyptus residues (11.4 wt % vs. 1.0 wt %). Similarly, the content of water extractives was much higher in wheat straw than in eucalyptus residues (9.4 wt % vs. 3.3 wt %). The observed differences are typical, as wheat straw is a form of herbaceous residue, whereas *Eucalyptus globulus* is an example of hardwood; the composition of both types of biomasses presented in this work are similar to those described in the literature [34,35,36,37].

### 3.2. Pre-Treatment with Imidazole

#### 3.2.1. Wheat Straw

All performed experiments resulted in two solid fractions composed mainly of polysaccharides (cellulose and hemicellulose). The macromolecular composition of cellulose produced in the pre-treated wheat straw, as well as the percentage of recovered solid fraction, are shown in Figure 2 and Appendix A.

It can be observed that at the mildest temperature studied (130 °C), the highest solid recovery was obtained, with a value of 73.7 ± 0.6 wt %. An increase in the reaction temperature led to a gradual decrease in solids recovery, achieving only 60.7 ± 0.7 wt % for the most severe reaction conditions (160 °C/4 h).

The obtained results also demonstrate that temperature is a dominant variable for cellulose recovery. An increase in temperature for the same reaction time, e.g., 2 h, enables enhancing the cellulose content by almost 15%, from 48.2 ± 0.1 wt % at 130 °C to 55.1 ± 0.1 wt % at 160 °C. On the other hand, an increase in the reaction time from 2 to 4 h, at the same temperature, showed no influence on the cellulose content in the pre-treated solids.

Regarding the hemicellulose content, an increase in either temperature or reaction time had no influence, changing the hemicellulose content by far less than 15%; e.g., for 130 °C/2 h, it was 25.6 ± 0.5 wt %, while for 160 °C/2 h, it was 24.2 ± 0.7 wt %.

The third main fraction of the pre-treated solid is lignin. Similarly, the lignin content was significantly affected by temperature. For example, for 2 h, an increase in the reaction temperature from 130 °C to 160 °C resulted in a reduction in lignin content from 9.6 ± 0.0 wt % to 5.3 ± 0.1 wt %. For a double-reaction time and the same range of reaction temperatures, the decrease was even more pronounced; it dropped by almost 60%, i.e., from 9.6 ± 2.4 wt % to 4.1 ± 0.0 wt %. 

The alkaline pre-treatment including imidazole disrupts the ester bonds between lignin and hemicellulose, and breaks the hydrogen bonds between lignin, cellulose, and hemicellulose, as demonstrated elsewhere [38]. This makes imidazole a good delignification agent [26,39]. The same finding was observed in the present work because the delignification yield, depending on the process conditions, varied from 59.6 ± 0.5 wt % to 85.9 ± 0.3 wt %. At the same time, the cellulose recovery was very high, being 85.4 ± 3.0 wt % at 160 °C/4 h, and, at 130 °C/2 h, it was 91.5 ± 1.8 wt %.

The obtained results are in agreement with the previous work presented in the literature. For example, Morais et al. [15] obtained 62.4 wt % of cellulose in the solid fraction with a lignin removal of 91.4 wt %, at 170 °C/2 h. This is a significant increase when compared to the results obtained at 110 °C/2 h, i.e., 42.2 wt % of cellulose content and 54.5 wt % of lignin removal. Furthermore, they also reported a reduction in solids and hemicellulose recovery with an increase in the reaction temperature. It is worth emphasizing that the temperatures used in the referred work were higher than those reported here. This may explain the differences in the obtained results. Toscan et al. [39] also reported that in the case of elephant grass, the cellulose content increased from 40.3 (114.4 °C/57 min) to 52.5 wt % (135.6 °C/308 min), whereas the lignin content dropped from 9.4 wt % at 125.0 °C/5.0 min to 4.6 wt % at 135.6 °C/308 min (increase in delignification from 50.7 to 81.8 wt %). Similarly to this work, they also reported a reduction in the solid recovery yield with an increase in temperature, with an 82.4 wt % yield for experiments performed at 114.4 °C/57 min, while for 140 °C/182.5 min the solid recovery yield was only 59.9 wt % [39].

Comparing the obtained results to those reported in the literature for different biomass pre-treatment technologies, (e.g., organosolv), it can be stated that the imidazole process is very effective for the delignification of biomass. Salapa et al. (2017) analyzed the organosolv pre-treatments of wheat straw [40]. They reported a production of a polysaccharide-rich fraction with 66.6 wt % of cellulose and 60 wt % lignin removals for pre-treatment with 50% (*v*/*v*) ethanol at 180 °C/20 min and 23 mM H_2_SO_4_. However, pre-treatment with 50% (*v*/*v*) acetone at 180 °C/40 min and 23 mM H_2_SO_4_ allowed to achieve 76.4 wt % delignification with 67.2 wt % cellulose. The same authors also studied the biomass processing at 160 °C. At this temperature, the maximum delignification achieved was 65 wt %, with acetone as the solvent, 23 mM H_2_SO_4_ and 40 min of reaction time. These results are considerably poorer than the delignification achieved in the present work at the same temperature. In addition, Wildschut et al. (2013) studied organosolv pre-treatment of wheat straw [41] and obtained a high-purity polysaccharide fraction with 75 wt % of cellulose and a delignification of 75 wt % at 190 °C/1 h 60% aqueous ethanol, with 30 mM H_2_SO_4_. Although these results seem to be better than those reported herein, it is important to emphasize that in both abovementioned works, the temperature used was much higher and additional external catalysts were used. Wildschut et al. (2013) studied the influence of temperature on pre-treatment without any catalyst and achieved a delignification of only 4.7 and 14.4 wt % for 160 and 170 °C, respectively [41]. It is also worth mentioning that novel organosolv pre-treatments have been studied at lower temperatures. Park et al. [42] pre-treated corn stover and achieved a 90.3 wt % delignification with a flow-through process including a 150 °C/60 min reaction with an aqueous solution of 30 wt % ethanol and 10 wt % H_2_O_2_. 

It is also important to compare the data obtained in the present work to those reported using ionic liquids (ILs). As a precursor of ILs, imidazole can also be considered a cheaper alternative to ILs in biomass processing [40]. Ren et al. [43] studied the pre-treatment of wheat straw with nine novel renewable cholinium-based ILs and reported a maximum delignification of 68.8 wt % with cholinium taurate under N_2_ stirred at 90 °C/6 h, which is a lower temperature than those applied herein. In their study, da Costa Lopes et al. reported a solid fraction with 83.4 wt % cellulose and only 2.8 wt % residual lignin, obtained with 1-butyl-3-methylimidazolium thiocyanate at 120 °C for 6 h with a 5 wt % biomass/IL ratio [44]. The same authors also studied biomass fractionation with 1-ethyl-3-methylimidazolium acetate, under the abovementioned conditions, and obtained carbohydrate-rich materials and a separated lignin fraction with 87 wt % purity [45]. Brandt et al. [46] tested a different 1-butyl-3-H-imidazolium hydrogen sulfate ionic liquids and found that this IL was able to remove up to 93 wt % of the lignin present in the raw material in a process carried out at 120 °C/20 h. Even though ILs allowed achieving similar results at similar temperatures to those used in this work, imidazole has the advantage of being a much cheaper reagent than most of the ILs tested so far.

#### 3.2.2. Eucalyptus Residues

To compare the pre-treatment efficiency of both biomasses, the same procedure was applied for the pre-treatment of eucalyptus residues. The macromolecular composition of solid fractions produced during pre-treatment, as well as the percentage of recovered solids from the initial biomass content in the respective fraction, are shown in Figure 3 and Appendix A. The solids recovery decreased with temperature, from 95.9 ± 0.1 wt % at 130 °C/4 h to 92.0 ± 0.7 wt % at 160 °C/4 h; however, the decrease was very low. The eucalyptus residues processing with imidazole under different conditions had almost no effect on the selectivity of this biomass fractionation. The solid fraction composition was almost constant for all the experimental conditions tested. The cellulose content changed from 50.3 ± 1.2 wt % to 54.3 ± 0.2 wt % for 130 °C/2 h and 160 °C/4 h, respectively. This means that in pre-treatment of eucalyptus residues with imidazole, cellulose was almost not affected because a quantitatively similar recovery of this macromolecule under all conditions tested was achieved.

When analyzing the lignin recovery, it can be stated that the pre-treatment of eucalyptus residues with imidazole has little, if any, apparent effect in experimental conditions on the lignin content. For all tested conditions, the lignin content varied from 17.1 ± 0.9 to 15.9 ± 0.4 wt %. Consequently, almost no change in the delignification yield was observed. In the range of studied conditions, the removal of lignin was between 52.6 wt % and 56.6 wt % for the least and the most severe reaction conditions, respectively.

Other studies of the pre-treatment of eucalyptus residues with acidic ionic liquids also demonstrated that delignification is very limited, and that cellulose can be recovered in quantitative amounts. Only in the switch to a hydrogen-bond alkaline ionic liquid, namely, 1-ethyl-3-methylimidazolium acetate, were the eucalyptus residues pre-treated more efficiently; as little as 0.7 or 1.0 wt % of lignin was found in the solid fractions produced at 120 °C/2 h and 140 °C/2 h [34].

#### 3.2.3. Comparison between Wheat Straw and Eucalyptus Residues Pre-Treatments

Both biomasses were subjected to the same pre-treatment procedure, where two variables (temperature and time) were studied. This protocol enables the comparison of the pre-treatment efficiency in two different feedstock origins: agriculture and forest. The efficiency was evaluated in terms of purity of the solid fraction and hemicellulose and lignin removal.

In general, the temperature had a similar effect on the solid yields, as with an increase in temperature, the solids recovery decreased for both biomasses. Regarding the solid fraction, the lowest cellulose content was achieved under the mildest conditions (48.2 ± 0.1 wt % at 130 °C), for wheat straw. By contrast, in terms of eucalyptus residues, the lowest cellulose content was achieved for pre-treatment with imidazole after 3 h at 145 °C (52.7 ± 0.4 wt %). The highest cellulose content was achieved for the most severe conditions used in this study, i.e., at 160 °C/4 h, and was similar for both biomasses – 55.1 ± 0.1 wt % and 54.3 ± 0.2 wt %, for wheat straw and eucalyptus residues, respectively. 

Analyzing the data presented above, it can also be concluded that for wheat straw, lignin and hemicellulose recoveries decrease with temperature, while the cellulose recovery showed negligible changes. Therefore, this difference indicates that the purity of cellulose in the solid fraction increases with temperature, and this conclusion is valid for both biomasses studied.

A significant difference between both biomasses was observed in terms of lignin removal. The lignin content in the solid fraction of wheat straw decreased considerably with temperature, while in the processed eucalyptus residues, the lignin content showed negligible changes under pre-treatment conditions. In addition, wheat straw pre-treatment reached delignification yields as high as 85.9 ± 0.3 wt % at 160 °C/4 h, indicating that imidazole is more efficient in the delignification of wheat straw than of eucalyptus residues, for which the delignification yield was only 56.6 ± 0.3 wt % under the same conditions. One of the reasons for this might be that hardwoods are more recalcitrant than herbaceous and agricultural residues, making their pre-treatment more challenging. This is probably attributed to their more rigid structure and higher lignin content, compared to the two others [47]. Most of the lignin in wood is bonded to hemicellulose components like a cementing agent, resulting in a complex and inaccessible structure. The differences observed in this work could also be due to variations in the lignin-carbohydrate association, the lignin distribution, or the lignin structure itself in agricultural residues and softwoods, which is mostly composed of guaiacyl units, while agricultural wastes contain not only guaiacyl but also syringyl and p-hydroxyphenyl units [48]. Thus, it can be concluded that although biomass fractionation with imidazole is possible, the process is less efficient with wood than with herbaceous biomass and others, as is also described in the literature [45,48,49].

Comparing the results of both biomasses pre-treated with imidazole to those presented in the literature for ionic liquid pre-treatment [34], some similarities can be noted. For example, the pre-treatment of both biomasses with 1-ethyl-3-methylimidazolium acetate at 120 °C/2 h produced solids with a very similar cellulose content, i.e., 65.8 wt % and 66.7 wt % for wheat straw and eucalyptus residues, respectively. Furthermore, the observed higher lignin content in processed eucalyptus residues was also found for the same ionic liquid. The lignin content in pre-treated eucalyptus residues was as high as 23.1 wt %, while for the pre-treated wheat straw, it was 17.2 wt % [34].

### 3.3. Enzymatic Hydrolysis

To evaluate the influence of a pre-treatment method on reducing sugars production, the native biomasses and recovered solids were subjected to enzymatic hydrolysis. The results are depicted in Figure 4 and Figure 5 and Appendix A for wheat straw and Figure 6 and Figure 7
Appendix A for eucalyptus residues.

In general, the glucan to glucose yield increased for all the samples obtained after pre-treatment with imidazole; however, the enzymatic digestibility of the polysaccharide was strongly dependent on the pre-treatment conditions and the biomass used. For both biomasses, an increase in the hydrolysis time resulted in an increase in either glucose or xylose released for all samples tested. 

Figure 4 and Figure 5 demonstrate that with an increase in the severity of pre-treatment conditions (temperature and time of reaction), an increase in the enzymatic digestibility of pre-treated wheat straw was observed. These results were expected since either temperature or time affect the composition of the recovered solids, as discussed above. A maximum glucan to glucose yield was achieved after 72 h of enzymatic hydrolysis for the solids subjected to the most severe pre-treatment conditions (160 °C/4 h), obtaining 93.3 ± 1.6 mol% and 77.3 ± 1.0 mol% for glucan to glucose and xylan to xylose yields, respectively. In a previous report [15], it was demonstrated that temperature played an important role in improving the enzymatic hydrolysis of solids obtained in the pre-treatment of wheat straw with imidazole. An increase in process temperature from 130 to 160 °C for 2 h pre-treatment time resulted in an increase in the glucose yield from 62.6 ± 1.5 to 84.3 ± 0.8 mol %, an increase by 193% and 294%, respectively, in comparison to native biomass. 

Pre-treatment time also affects enzymatic hydrolysis and an increase in the reaction time leads to an increase in the glucan to glucose yield. However, in this case, the difference is less pronounced. For both temperatures, 130 °C and 160 °C, an increase in pre-treatment time from 2 to 4 h enabled an increase in the glucan to glucose yield by 11.6% and 10.8%, respectively. Hence, in the case of wheat straw, the main factor influencing the enzymatic hydrolysis is the pre-treatment temperature.

The enzymatic digestibility of polysaccharide fractions is strongly dependent on several factors and one of the main factors influencing the enzymatic hydrolysis is the lignin content. The obtained results demonstrated that a linear relation exists (Glucan to glucose yield (mol%)=122.97*delignification (% (w/w)−9.465, R2=0.94) between the biomass delignification and the efficiency of enzymatic digestibility. The extensive removal of lignin favors enzymatic hydrolysis, confirming that lignin is one of the most important inhibitors of efficient enzymatic hydrolysis. 

Analogous to wheat straw, the pre-treatment of eucalyptus residues with imidazole improved the enzymatic hydrolysis, especially when compared to the native biomass, as demonstrated in Figure 6 and Figure 7. However, in this case, the yield of enzymatic hydrolysis was almost independent of pre-treatment conditions, obtaining 40 mol% for the glucan to glucose yield after 72 h of enzymatic hydrolysis for all conditions tested. Even an increase in hydrolysis time to 144 h allowed only a small increase in the glucan to glucose yield, to merely 47.6 ± 0.9%.

Although imidazole pre-treatment resulted in a four-fold increase in glucose yields in comparison to native biomass, the hydrolysis yields detected were lower than those observed for wheat straw (2.3-fold lower). This difference again confirms that the biomass type is crucial in efficient biomass processing for obtaining a high enzymatic hydrolysis yield. Similar conclusions were drawn in the case of ionic liquid pre-treatment with the same biomasses [34], where it was demonstrated that, besides the lignin removal, a change in cellulose crystallinity is fundamental to achieve the efficient release of reducing sugar.

## 4. Conclusions

The performed studies demonstrated that imidazole can be considered a potential solvent for biomass pre-treatment. Imidazole allowed us to separate the main fractions of wheat straw and to produce pre-treated solids. Such solids were shown to be suitable feedstock for reducing sugar production. On the other hand, eucalyptus residues were demonstrated to be much more resistant to imidazole processing; very limited delignification of this residue was achieved in the range of studied conditions. This, in turn, resulted in a very low enzymatic hydrolysis yield. 

Therefore, imidazole was shown to be efficient for the treatment of herbaceous residues, whereas for hardwood, within the range of studied conditions, imidazole was not the best choice. These results confirm that the efficiency of pre-treatment, as well as subsequent enzymatic hydrolysis yield, are dictated by the reaction solvent and the properties of the processed biomass.

## Figures and Tables

**Figure 1 molecules-26-07591-f001:**
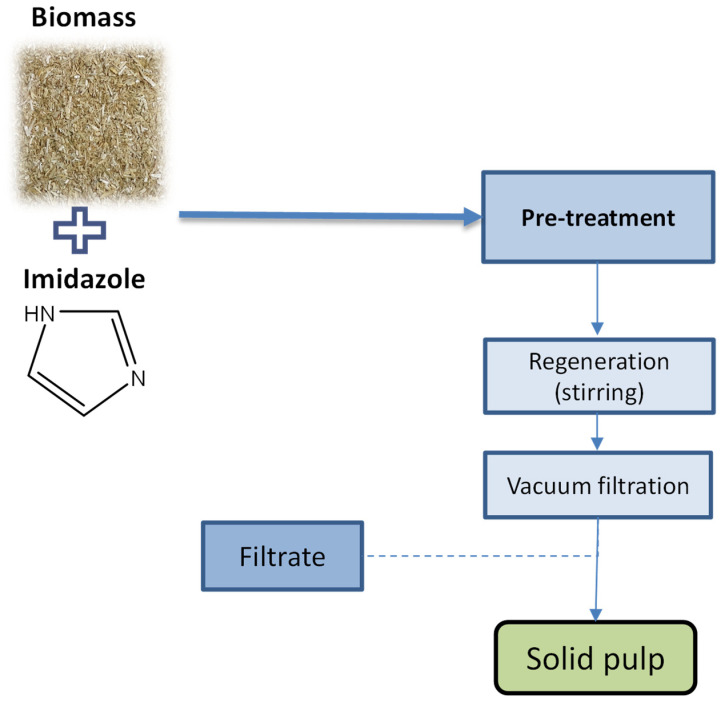
The procedure of biomass pre-treatment with imidazole.

**Figure 2 molecules-26-07591-f002:**
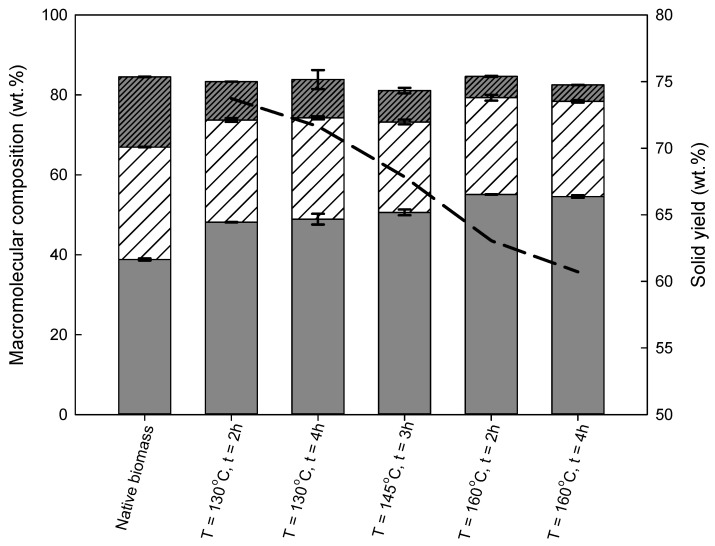
Native wheat straw and solid fraction compositions (wt %), obtained from pre-treatment with imidazole at varied reaction temperatures and times. The black line represents the recovered solids (wt %). The cellulose amount, depicted as a grey bar, was measured in terms of glucan content, and hemicellulose, represented as a white bar with a pattern, was measured as the sum of the xylan and arabinosyl group content. Lignin is depicted as a grey bar with a pattern. Ash and other minor components are not shown, for clarity in the figure.

**Figure 3 molecules-26-07591-f003:**
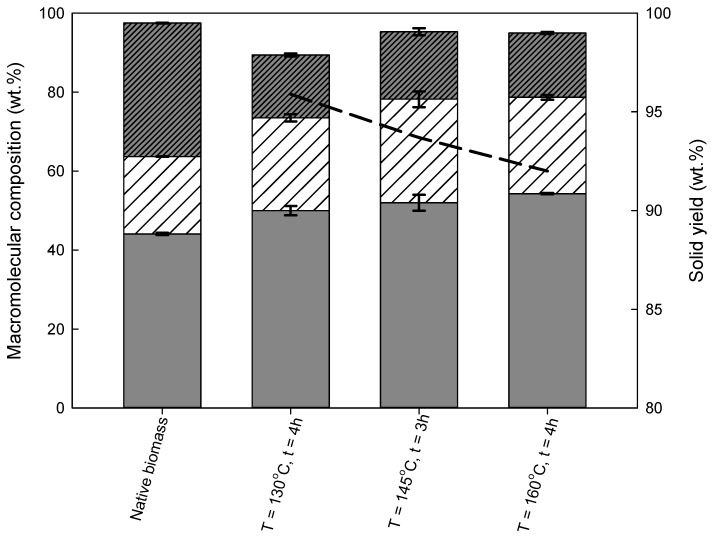
Native eucalyptus residues and solid fraction compositions (wt %), obtained from pre-treatment with imidazole at varied reaction temperatures and times. The black line represents the recovered solids (wt %). The cellulose amount depicted as the grey bar was measured as the glucan content. The hemicellulose amount represented as the white bar with a pattern was measured as the sum of xylan and arabinosyl group contents. Lignin content is depicted as the grey bar with a pattern. The contents of ash and other minor components are not shown for the clarity of the figure.

**Figure 4 molecules-26-07591-f004:**
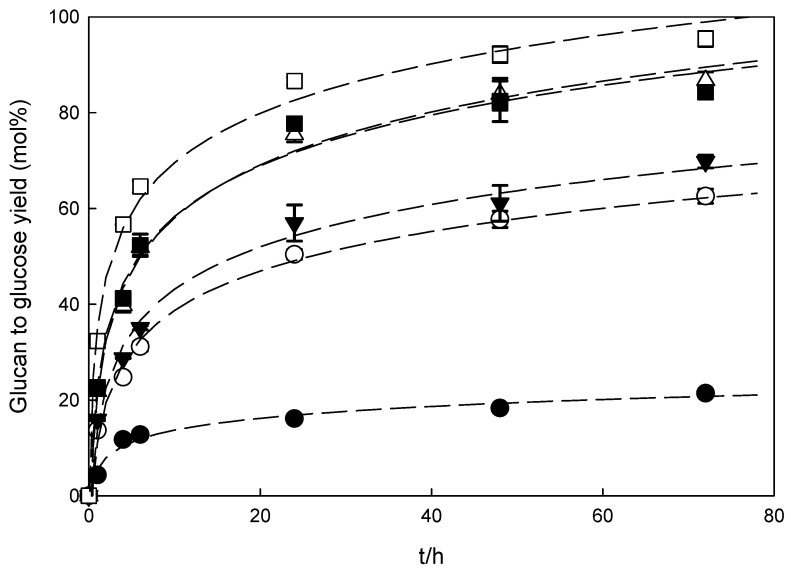
Glucan to glucose yields of wheat straw pre-treated solids that were produced by imidazole pre-treatment. The enzymatic hydrolyses were performed at 2% (*w*/*v*) total solids with 60 FPU/g glucan of Cellic CTec2. The dashed lines are only used as a guide for the eye and have no physical meaning.

**Figure 5 molecules-26-07591-f005:**
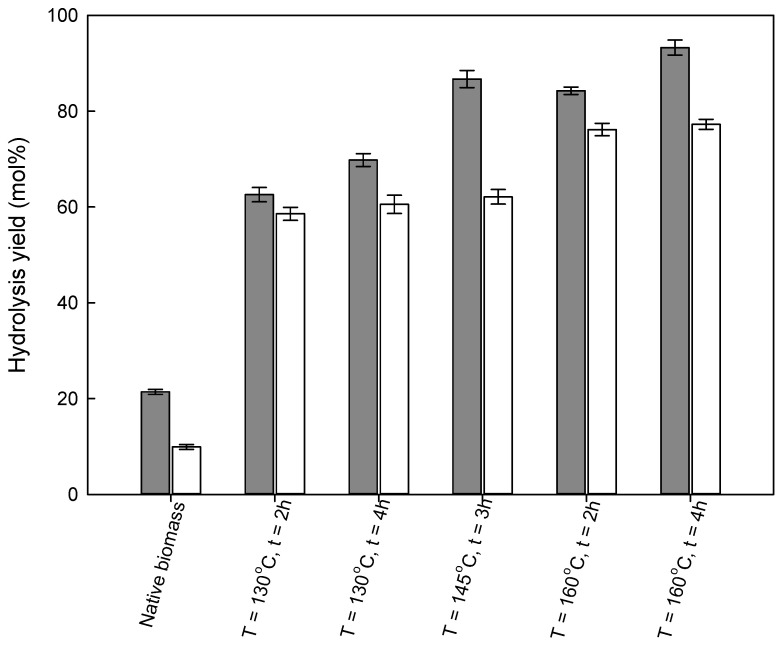
Glucan to glucose (grey bar) and xylan to xylose saccharification (white bar) yields, over 72 h of enzymatic hydrolysis time of wheat straw pre-treated solids, as a function of the pre-treatment conditions. The enzymatic hydrolysis yields for native wheat straw are presented for comparison.

**Figure 6 molecules-26-07591-f006:**
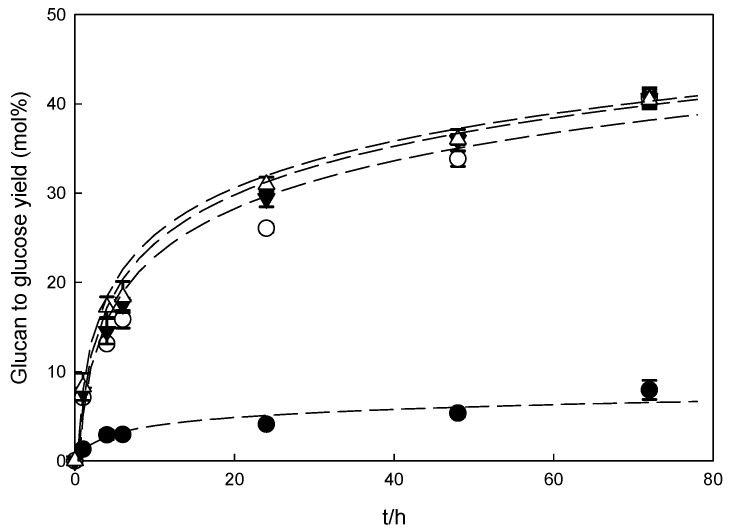
Glucan to glucose yields that were produced by an imidazole pre-treatment of eucalyptus residues. The enzymatic hydrolyses were performed at 2% (*w*/*v*) total solids with 60 FPU/g glucan of Cellic CTec2. The dashed lines are only used as a guide for the eye and have no physical meaning.

**Figure 7 molecules-26-07591-f007:**
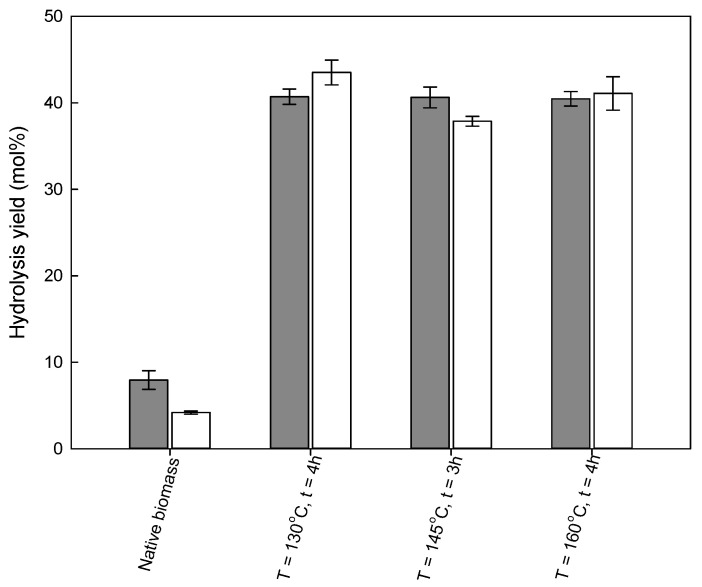
Glucan to glucose (grey bar) and xylan to xylose saccharification (white bar) yields for 72 h of enzymatic hydrolysis of eucalyptus residues as a function of the pre-treatment conditions. The enzymatic hydrolysis yields for native wheat straw are presented for comparison.

**Table 1 molecules-26-07591-t001:** Feedstock composition (wheat straw and eucalyptus residues).

Components (Dry Weight %)	Wheat Straw	Eucalyptus Residues
Glucan	35.9 ± 0.3	44.1 ± 0.9
Hemicellulose	26.7	19.6
Xylan	22.1 ± 0.6	15.7 ± 0.2
Arabinosyl group	2.0 ± 0.7	0.5 ± 0.1
Acetyl group	2.6 ± 0.9	3.4 ± 0.9
Lignin	16.7	33.8
Acid-insoluble	15.5 ± 0.4	26.4 ± 0.1
Acid-soluble	1.2 ± 0.1	7.4 ± 0.1
Ash	11.4 ± 0.1	1.0 ± 0.1
Extractives		
Water	9.4 ± 1.3	3.3 ± 0.4
Ethanol	1.4 ± 0.1	1.5 ± 0.1

## Data Availability

The data presented in this study are available in Appendix A.

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
