# Peer review of "Imidazole Processing of Wheat Straw and Eucalyptus Residues—Comparison of Pre-Treatment Conditions and Their Influence on Enzymatic Hydrolysis"

_molecules, 2021, doi:10.3390/molecules26247591_

Round 1

Reviewer 1 Report

The reviewed article is a useful study on lignin/carbohydrate extraction, a worthy addition to the array of available methods. The idea (using imidazole as a solvent) is not particularly innovative, but the application to representative grass and wooden biomass makes it worth publishing. The manuscript is well written, with a logical, clear presentation. The applied experimental techniques are sound and the design of conducted experiments is adequate.

Though well written, the manuscript needs several rounds of thorough editing. Without this treatment, it may have a lower impact because some passages are a bit hard to read and several statements are unclear to the point that they read ambiguous. I have made enough suggestions in the attached scanned file to make it the first round of editing. I suggest that the authors make the second iteration and then show their manuscript to a native English-speaking colleague. To detect any remaining awkward statements. The three areas in which the authors should seek improvement are:

  • Whenever you say “this,” add the corresponding noun.
  • Assure the consistency of verb tenses. Usually, past tense is used except for the statements showing the action clearly occurring at present, like “lignin and cellulose are the most abundant biopolymers.”
  • Work on the use of article "the"

Author Response

Attached file contains the answer to the Reviewer 1 comments.

Reviewer 2 Report

In the descriptive section of the material, I recommend that the authors read and supplement the manuscript with a discussion with research about more physical or energetic aspects of wood. I recommend reading the article: https://doi.org/10.3390/en14113270

In these articles, you can also find information about correlation using ANOVA with the Duncan test. Similar studies, in particular the division into homogeneous groups, were also missing in the article. You can find information about a fractional breakdown or CHONS analysis as well as correlation using ANOVA with the Duncan test. It will certainly enrich the manuscript.

The article contains statistical data, but they are at the primary level. The statistical analysis of the factors was not supported by any post-hoc test (eg Duncan), which would allow any more advanced conclusions to be drawn. The manuscript has very mediocre results.

Author Response

Attached file contains the answer to the Reviewer 2 comments.

Round 2

Reviewer 2 Report

Authors correct all comments.